# Acquisition, transmission and strain diversity of human gut-colonizing crAss-like phages

Benjamin A. Siranosian[1], Fiona B. Tamburini[1], Gavin Sherlock [1] & Ami S. Bhatt [1,2]*

CrAss-like phages are double-stranded DNA viruses that are prevalent in human gut microbiomes. Here, we analyze gut metagenomic data from mother-infant pairs and patients undergoing fecal microbiota transplantation to evaluate the patterns of acquisition, transmission and strain diversity of crAss-like phages. We find that crAss-like phages are rarely detected at birth but are increasingly prevalent in the infant microbiome after one month of life. We observe nearly identical genomes in 50% of cases where the same crAss-like clade is detected in both the mother and the infant, suggesting vertical transmission. In cases of putative transmission of prototypical crAssphage (p-crAssphage), we find that a subset of strains present in the mother are detected in the infant, and that strain diversity in infants increases with time. Putative tail fiber proteins are enriched for nonsynonymous strain variation compared to other genes, suggesting a potential evolutionary benefit to maintaining strain diversity in specific genes. Finally, we show that p-crAssphage can be acquired through fecal microbiota transplantation.

[1] Department of Genetics, Stanford University, Stanford, CA, USA. [2] Department of Medicine, Division of Hematology, Stanford University, Stanford, CA, USA.
*email: asbhatt@stanford.edu

In addition to trillions of bacteria, the human gastrointestinal tract is densely populated with bacteriophages. Bacteriophages can drive bacterial community composition and mediate horizontal gene transfer[1], and alterations in the human gut virome have been associated with disease[2,3]. However, our knowledge of the contributions of specific bacteriophages to human biology is limited, in part due to the paucity of viral sequences represented in reference databases. High-throughput sequencing and advanced genomic tools have facilitated the in silico discovery and characterization of previously unknown bacteriophages. The preeminent example of such a discovery is crAssphage (cross-Assembly phage), initially identified from human virome sequencing data[4]. A bacteriophage with a ~97 kilobase circular, double-stranded DNA genome, crAssphage sequences are found almost exclusively in human fecal metagenomes in diverse populations globally[5–8], and can be highly abundant. Initial estimates indicate that crAssphage is present in up to 73–77% of humans[4,5]. Given the near ubiquity of crAssphage and its apparent specificity to the human gut, quantitative PCR assays have been developed to use crAssphage genes as markers for tracking human fecal pollution in water and environmental samples[9,10] and in human stool[11].

More recent investigations have shown that crAssphage is one member of a wide range of crAss-like phages that exist in the human microbiome[5,12]. In this manuscript, we adopt the taxonomic classification system for crAss-like phages used in Guerin et al.[5], which proposed 4 subfamily (Alpha, Beta, Gamma, Delta) and 10 cluster (1–10) designations based on shared protein coding genes. The crAssphage first described by Dutilh et al.[4] belongs to the Alpha subfamily, cluster 1 and is given the designation prototypical crAssphage (p-crAssphage); we use p-crAssphage in all further designations to avoid ambiguity. The genomes classified as crAss-like phages by Guerin et al.[5] are diverse - members of the same cluster share at least 40% of protein coding genes, while members of the same subfamily share only 20–40% of protein coding genes.

It is not known whether or how crAss-like phages influence host biology or disease[13,14]. To answer higher-order questions about the role of crAss-like phages in human biology, it is necessary to establish basic principles of acquisition, persistence, and distribution. Although p-crAssphage has been detected in infant gut metagenomes[7,13], it is not yet known how crAss-like phages are acquired in infancy. Infants acquire many of their first microbes, such as *Bacteroides* species, from their mother during and after delivery[15–18]. By contrast, it has been demonstrated that adult twins and their mothers have unique gut viromes[19]. Given that *Bacteroides* species are hypothesized to be the bacterial host(s) of p-crAssphage[4,20], and the apparent specificity of p-crAssphage to the human gut as opposed to other mammals or environmental samples, we postulated that p-crAssphage is vertically transmitted from mother to infant, similar to what is observed for many bacterial taxa and in contrast to what is reported for other members of the human virome. To test this hypothesis, we examined publicly available shotgun metagenomic data from two stool microbiome datasets[15,16] consisting of samples from mothers and their infants ($n = 143$ mother–infant pairs).

In this study, we find that p-crAssphage and other crAss-like phages are rarely detected in the gut microbiome at birth but become detectable during the first year of life. We observe >99.5% identical genome sequences in one half of cases where mothers and infants have the same crAss-like phage, suggesting vertical transmission from mother to infant. Infants acquire a reduced diversity population of p-crAssphage compared to their mother, but strain diversity expands upon colonization. Finally, by examining shotgun metagenomic data from patients undergoing fecal microbiota transplantation (FMT), we show that FMT recipients can acquire p-crAssphage with a nearly identical genome sequence as the stool donor. These results begin to uncover the principles of acquisition and transmission of p-crAssphage and other crAss-like phages, which are the most prevalent human-associated phages described, to date.

## Results

### Presence of p-crAssphage in mother and infant microbiomes.
We evaluated the presence and abundance of p-crAssphage in the microbiomes of mothers and infants by classifying sequencing reads with Kraken2[21], using a database of all bacterial, viral and fungal genomes in NCBI GenBank assembled to complete genome, chromosome or scaffold quality as of February 2019 (see Methods). P-crAssphage is represented by a 97-kb genome (accession NC_024711.1). Assigning absolute presence or absence of an organism in metagenomic sequencing data is difficult and confounded by sequencing depth. Here, we consider samples with ≥1,000 reads classified as p-crAssphage to be evidence for presence, as this corresponds to roughly 1x coverage of the genome (assuming 100 bp reads and a ~100-kb genome length). Samples from mothers and infants had an average of 8.7 M reads after preprocessing, and the 1000 read coverage threshold thus corresponds to an average relative abundance of 0.011%. Of note, this somewhat arbitrary threshold, while fairly specific, renders our approach limited in sensitivity - that is, we do not report on p-crAssphage when it is present at lower relative abundance.

Although p-crAssphage is highly abundant in the adult gut microbiome[5,14] and has been detected in infant gut microbiomes, it is unclear when or how it is acquired. Consistent with previous studies describing low relative abundance or absence of p-crAssphage in the infant gut microbiome[7,22], we found 0 out of 22 infants have ≥1000 p-crAssphage reads in samples collected within 24 h of birth (Supplementary Data 1). P-crAssphage increases in prevalence as infants age: it is detected in 3/35 (9%) infants from Yassour et al.[16] by three months and 16/100 (16%) infants in Bäckhed et al.[15] by 12 months. P-crAssphage is more prevalent in adult mothers, where it is detected in at least one sample from 8/35 (23%) and 25/100 (25%) of mothers in each study, respectively (Supplementary Fig. 1). P-crAssphage is detected in both the mother and her infant in ten cases, while 23/33 p-crAssphage positive mothers have a p-crAssphage negative infant, and 9/19 p-crAssphage positive infants have a p-crAssphage negative mother.

The infant gut microbiome is strongly impacted by delivery mode, and it has been shown that infants born by Cesarean section initially lack *Bacteroides* species[16,18,23–25]. In line with the described effects of Cesarean section delivery and a hypothesized Bacteroidetes host, we found that all 19 p-crAssphage positive infants were delivered vaginally, while all 21 infants delivered through Cesarean section remained p-crAssphage negative. Although not significant when each study is tested individually, the association between delivery mode and p-crAssphage presence is significant when samples from the two studies are combined ($p = 0.043$, Fisher's exact test). This result contrasts with the findings of McCann et al.[7], where the authors found no association between p-crAssphage relative abundance and delivery mode.

### Putative vertical transmission of p-crAssphage.
Vertical transmission of gut microbes from mother to infant is common and well-described among certain bacterial taxa[15–17,26]. To test the hypothesis that p-crAssphage can be vertically transmitted from mother to infant, we investigated metagenome-assembled p-crAssphage genomes from ten p-crAssphage positive mother-infant pairs (see Methods). In six cases, mother-infant pairs had nearly identical

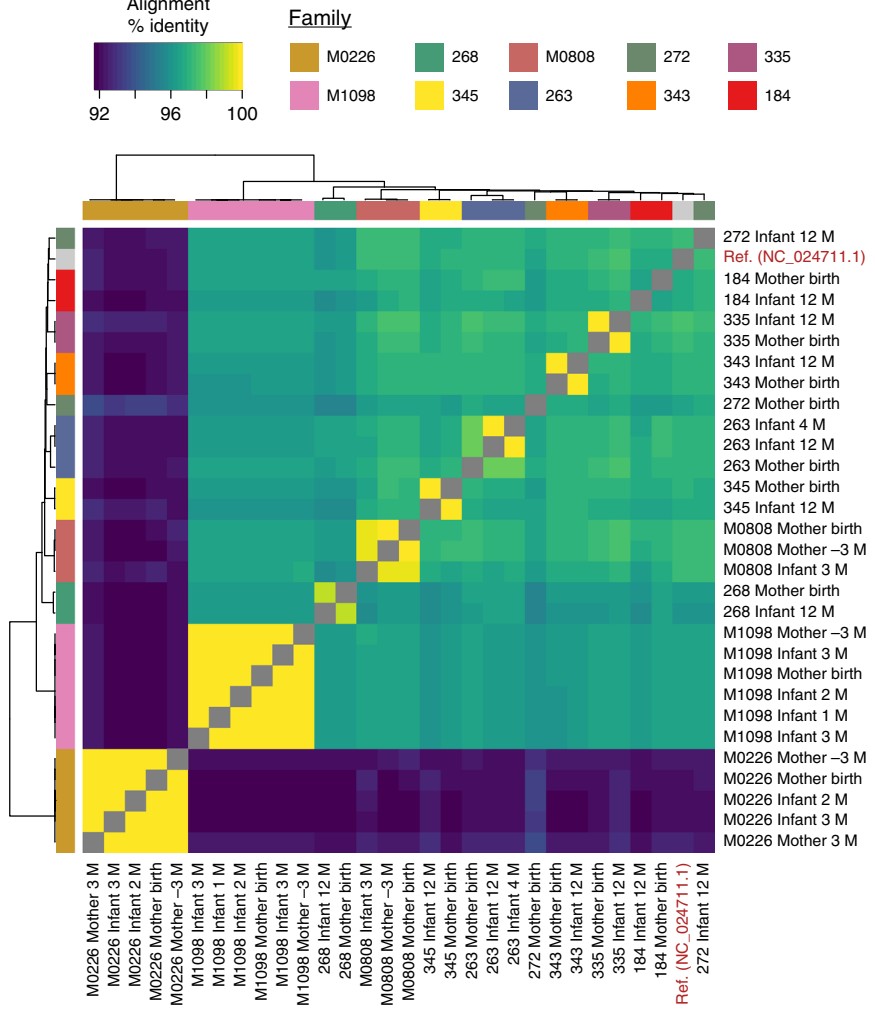

**Fig. 1 Mother-infant pairs share > 99.7% similar p-crAssphage genomes in 6/10 cases.** Heatmap of pairwise alignment percentage identity of metagenome-assembled p-crAssphage genomes from mothers and infants. Only families with p-crAssphage detected in at least one mother and infant sample are shown. The p-crAssphage reference genome is also included as a comparison.

assembled sequences (families M0226, M0808, M1098, 335, 343, and 345; >99.7% similarity). Two mother-infant pairs had assembled sequences more similar than unrelated pairs (families 263 and 268; 98–99.2% similarity), and two mother-infant pairs had assembled sequences that were no more similar than unrelated pairs (families 184 and 272; ~96% similarity) (Fig. 1). Overall, related mothers and infants harbor more closely related p-crAssphage sequences than unrelated mothers and infants (Supplementary Fig. 2a). When all samples with sufficient p-crAssphage coverage were included in the assembly comparison, no pairs from unrelated individuals had >98% similarity (Supplementary Fig. 3, Supplementary Data 3). Assembled p-crAssphage genomes were high quality and contiguous: in the 29 samples from families with p-crAssphage found in mothers and infants, the median contig N50 was 59.6 kb (standard deviation, SD = 41.4 kb), median number of contigs was 3 (SD = 17) and median total assembled length was 96.3 kb (SD = 24.2 kb) (assembly statistics reported in Supplementary Data 2). One-to-one pairwise alignments between assembled sequences from mother-infant pairs had a median length of 85.8 kb (SD = 27.8 kb). In all, 22 samples assembled a nearly complete (>95 kb) p-crAssphage genome in a single contig. The assembled genomes also share 91.2–97.6% nucleotide identity with the p-crAssphage reference genome, adding confidence that they are truly representative assemblies. Next, we used a variant calling approach to identify fixed SNPs compared to the

p-crAssphage reference (see Methods). Pairs of samples were compared at genomic sites covered ≥10x and used to construct a heatmap of SNP similarity. The same six mother-infant pairs (M0226, M0808, M1098, 335, 343, and 345) had >99.5% SNP similarity and continued to cluster together (Supplementary Fig. 4). Mother-infant pairs had higher SNP identity than unrelated pairs on average (Supplementary Fig. 2b).

**Strain diversity in the p-crAssphage population.** Metagenomic assembly only represents the dominant allele at each position, and a fixed SNP comparison only considers sites that are identical across all strains present in a sample. To understand the differing p-crAssphage strains present in the microbiome, we would ideally phase strain "haplotypes" with a technology like long-read sequencing. With only short reads available, we examined genomic positions that had multiple single nucleotide variant alleles called at high-quality (≥5 reads for each allele, multiallelic sites) as a proxy for strain diversity. We report a normalized statistic ($F_{multi}$) to compare multiallelic sites across samples with highly variable coverage. At a given minor allele fraction (AF), $F_{multi}$ is the proportion of multiallelic sites with a minor AF > $x$ among those sites covered well enough to detect a minor AF of $x$.

We calculated $F_{multi}$ for minor allele fractions of 0.40, 0.30, 0.20, and 0.10 and compared across samples at a given minor AF

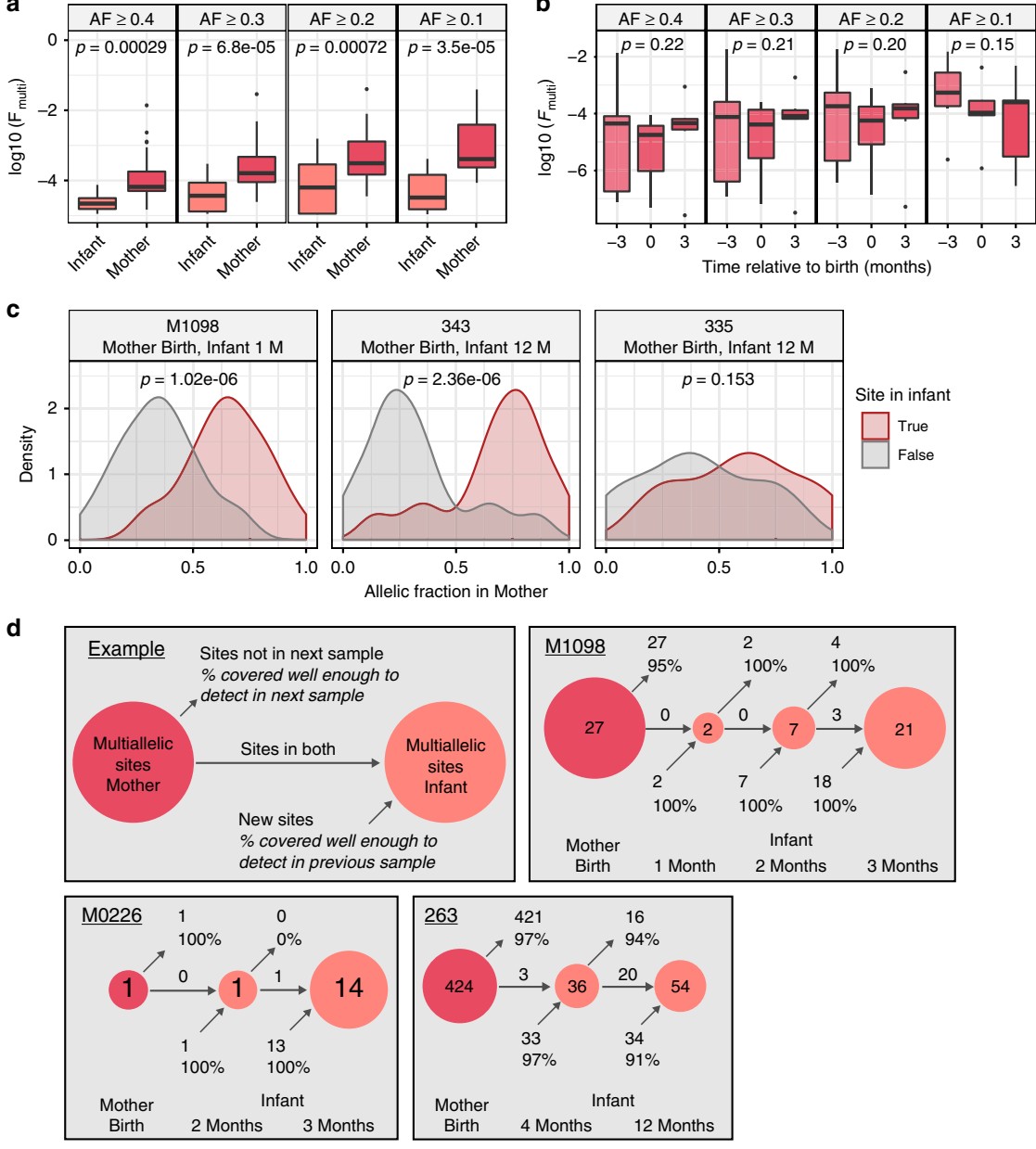

**Fig. 2 P-crAssphage populations in mothers and infants differ in strain diversity. a** The p-crAssphage population in mothers has more multiallelic sites than the p-crAssphage population in infants. $F_{multi}$ (fraction of the p-crAssphage genome with multiallelic sites detected at the given allelic fraction threshold) in all mother and infant samples with at least one multiallelic site detected. P-values were calculated with the two-sided Wilcoxon rank-sum test and are uncorrected for multiple hypothesis testing. **b** P-crAssphage populations in mothers do not change in the number of multiallelic sites over time. $F_{multi}$ for mother samples from Yassour et al.[16]. P-values were calculated with a linear mixed model to account for repeated sampling of the same individual. **c** Allelic fraction of multiallelic sites in the p-crAssphage genome from mothers that are fixed in her infant. The distribution is separated by alleles that are present in the infant's p-crAssphage or not. P-values were calculated with the two-sided Wilcoxon rank-sum test. **d** Schematic depicting multiallelic sites in mother and infant samples over time. In the three cases where p-crAssphage was detected in the mother and multiple samples from the same infant, infants develop more multiallelic sites over time. Additional examples of multiallelic sites in mothers and infants are provided in Supplementary Fig. 10. In all boxplots, boxes extend to the first and third quartile, whiskers extend to the upper and lower value within 1.5* interquartile range (IQR) from the box. Outliers are shown as points.

value. At every AF tested, infants as a group had a smaller fraction of multiallelic sites when compared to mothers as a group. (Fig. 2a). In the three cases where we detected p-crAssphage in multiple samples from the same infant, we found more multiallelic sites in later samples. We were typically powered enough to detect multiallelic sites of the observed AF in earlier samples, but we cannot rule out the possibility that newly observed variants are below our limit of detection in earlier

samples. Multiallelic sites in infants are often fixed sites in the p-crAssphage population of the mother (Fig. 2d, Supplemental Fig. 10). In contrast to infants, mothers from Yassour et al.[16] showed no change in the proportion of multiallelic sites over the 6 month sampling period (Fig. 2b). We then looked at multiallelic sites in mothers that are fixed in matched infant samples. In 2/3 cases where we observed putative p-crAssphage transmission and the mother had ≥10 multiallelic sites, major alleles are

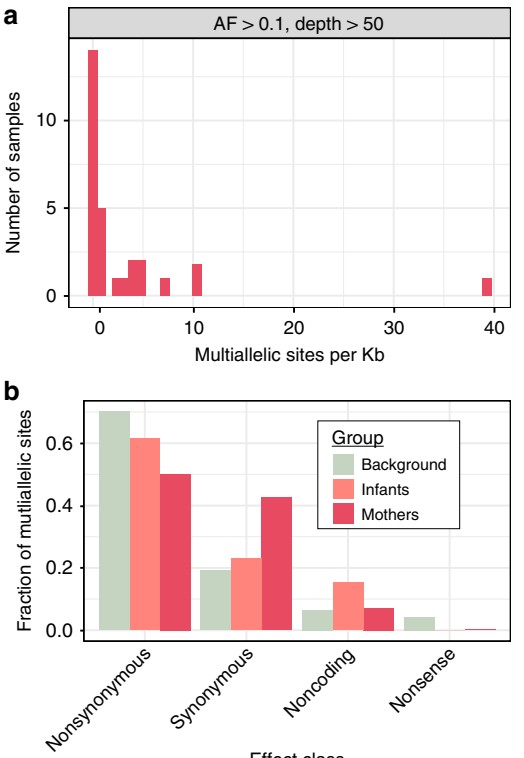

**Fig. 3 Predicted effects of multiallelic sites differ in the p-crAssphage population of mothers and infants. a** Distribution of multiallelic sites per kilobase in samples from mothers. **b** Distribution of predicted effects of multiallelic sites from mother and infant samples, compared to a background distribution of equal probability of each DNA change at each position in the p-crAssphage reference genome.

disproportionately detected in the child (Fig. 2c). Taken together, these results suggest a few potential models: one is that infants acquire a single strain or limited diversity of p-crAssphage strains. As the infant microbiome matures and diversifies with age, the p-crAssphage population can evolve and acquire genetic diversity. Alternatively, a larger spectrum of p-crAssphage strains than is detected may be harbored in a mother, with some strains below the limit of our detection. These strains may experience less selective pressure in the infant than in the mother; thus, these strains may be more numerous and easily detected in the infant than they are in the mother.

We continued to use multiallelic sites to investigate p-crAssphage strain diversity within an individual. Mothers generally have limited strain diversity, with a median of 0.41 (SD = 7.5) multiallelic sites per kb at AF > 0.1; variation in 0.04% of the genome. In a small number of cases, we do observe samples with up to 100x more frequent multiallelic sites (Fig. 3a). This allelic variation could be the result of many closely related strains existing together or a smaller number of more divergent strains. Phasing strain "haplotypes" is necessary to distinguish between these possibilities. Limited strain diversity suggest an exclusion principle, which favors mono- or oligo-colonization of a particular p-crAssphage strain or closely related strains within the gut of an individual, though notably, a minority of individuals may be simultaneously colonized by multiple diverse strains. P-crAssphage has fewer multiallelic sites on average compared to Lactococcus phages, which are the only other group of phages detected at ≥1x coverage in at least ten samples. Using the reference genome of the most frequently detected individual phage, *Lactococcus phage 16802*, 34 samples had at least 1x

coverage; these samples had a median of 58.8 (SD = 14.7) multiallelic sites per kb at AF > 0.1 (Supplementary Fig. 9). We did not find assembled Lactococcus phage genomes with >99% similarity between any samples from different individuals.

We next evaluated whether strain variation in the p-crAssphage population was the result of synonymous or nonsynonymous genomic changes. Variant effects were predicted using the p-crAssphage genome annotation from GenBank and SnpEff[27] (see Methods). We compared the proportion of observed variant effects to a null model of equal probability of mutation at every base in the reference genome. In samples from mothers, multiallelic sites with predicted nonsynonymous and nonsense effects were less likely than expected under the null model, while synonymous sites were more likely than expected ($p < 1e-5$ for each category, likelihood ratio test; Fig. 3b). An overrepresentation of synonymous variants suggests that strain diversity in the p-crAssphage population of mothers is enriched for neutral genetic variation, which may have been acquired over the relatively long time the phages could have been present in the microbiome. In contrast to mothers, the predicted effects of multiallelic sites in infants were indistinguishable from the null model ($p > 0.05$ for each effect category, likelihood ratio test). This may suggest that multiallelic sites in infants arise randomly and the forces acting to influence the distribution of predicted sites in mothers have not had time to act on the infant's p-crAssphage population yet. Alternatively, selective pressures acting on p-crAssphage alleles may be entirely different in the infant and mother microbiomes. Comparing mother and infant distributions showed that only the proportion of synonymous multiallelic sites was significantly different between the two ($p = 0.04$, likelihood ratio test). We note that synonymous variants may not be truly neutral, as noncoding variants have been shown to affect translation efficiency in bacteriophages[28].

In adults, we find wide variation in the number of multiallelic sites across the p-crAssphage genome, with enrichment in the number of sites and the ratio of nonsynonymous to synonymous variants corresponding to certain predicted genes (Fig. 4). Multiallelic sites were detected in 80/88 predicted genes. When genes were ranked by the length-normalized number of nonsynonymous variants, "putative Tail sheath protein" was the top annotated gene, and other predicted tail proteins also had high ratios (Supplementary Data 4). Phage tail proteins are responsible for host tropism;[29–31] therefore maintaining multiple functionally different alleles in the population may be beneficial to expand the host range of p-crAssphage. Increased variation in tail fiber genes was also found in an analysis of p-crAssphage genomes from South Africa[32]. The genes that were least likely to have nonsynonymous multiallelic sites appear to be those that are critical for phage function, such as "putative portal protein", "putative major capsid protein" and putative RNA polymerase subunits. Interestingly, some of these genes correspond to peaks in the number of multiallelic sites detected, even though the variants had mostly synonymous predicted effects. Infant samples have fewer multiallelic sites than mothers, with sites detected in 33/88 genes. "Putative ssb single stranded DNA-binding protein" was the most enriched gene for nonsynonymous multiallelic sites. Five out of ten tail fiber proteins had at least one nonsynonymous multiallelic variant in infant samples, and the most frequently mutated tail fiber gene was "putative phage tail-collar fiber protein (*DUF3751*)," a top hit in the adult samples (Supplementary Data 4).

**Acquisition and transmission of crAss-like phages**. P-crAssphage is the first described member of an expanding group of crAss-like phages. Guerin et al.[5] assembled 249 complete or near-complete crAss-like phage genomes out of metagenomic sequencing datasets,

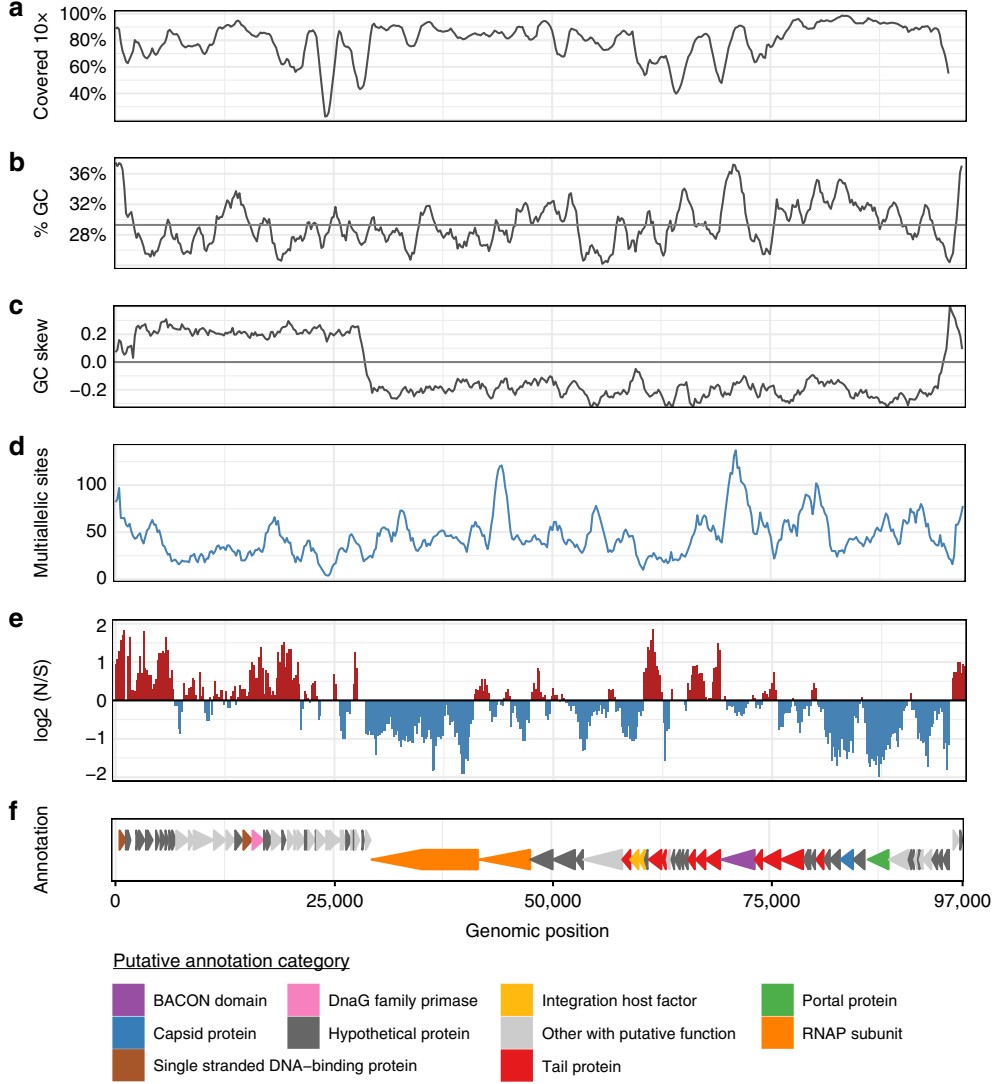

**Fig. 4 The frequency and predicted effects of multiallelic sites vary across the p-crAssphage genome. a** Fraction of samples from mothers covered at least 10x. All values are calculated with a sliding window of size 1500 bp with step size 200. **b** %GC content of the p-crAssphage reference genome. **c** GC skew of the p-crAssphage reference genome. **d** Total count of multiallelic sites (AF > 0.1) in the window. **e** Log base 2 ratio of nonsynonymous (N) to synonymous (S) multiallelic sites (AF > 0.1). **f** Annotation and selected predicted functions of genes in the reference genome.

which were then classified into four subfamilies (Alpha, Beta, Gamma, Delta) and 10 clusters (1–10) based on shared protein coding genes. P-crAssphage is a member of cluster Alpha 01. Given the observed sharing of p-crAssphage genome sequences by mother-infant pairs, we were interested to determine if similar putative transmission events could also be observed for crAss-like phages. We added the crAss-like genomes from Guerin et al.[5] to the Kraken2 viral reference database in a hierarchy following the proposed subfamily and cluster designations (see Methods). For classification and transmission analyses, we carried out analyses at the level of crAss-like phage clusters. A threshold of 1000 reads classified to the same cluster (roughy 1x coverage) was treated as evidence for presence.

Broadly, crAss-like phages are more frequently detected in the microbiome of mothers and infants than p-crAssphage alone. In total, 7/36 (19%) infants from Yassour et al.[16] and 49/100 (49%) infants from Bäckhed et al.[15] have at least one crAss-like cluster detected in at least one sample. At least one cluster was detected in 33/43 (77%) and 88/100 (88%) of mothers from each study. Mothers are most likely to be colonized by a single crAss-like phage cluster, although we observe samples with up to 8 clusters

detected (Supplementary Fig. 5). We do not observe any crAss-like phage cluster present in infant samples taken within 24 h of birth. However, two infants from Bäckhed et al.[15] have a crAss-like phage meeting the presence threshold in samples collected as soon as 3 days after birth (samples 385_B and 633_B). This may represent the lower time limit for the crAss-like phage and its bacterial hosts(s) to reach the detection threshold.

The putative hosts of crAss-like phages, members of the Bacteroidetes phylum, are known to be vertically transmitted[15–17,26]. Thus, we hypothesized that crAss-like phages would be more frequently transmitted to vaginally vs. Cesearan section born infants. We observed that all 19 p-crAssphage positive infants were delivered vaginally, while all Cesarean section born infants remained p-crAssphage negative for the duration of sampling. Although low sample numbers prevented this association from raising to the level of significance when each cohort was tested individually, it was significant when samples from both cohorts were considered together (Bäckhed $p = 0.12$, Yassour $p = 1$, combined $p = 0.043$, Fisher's exact test). We also tested for associations in cases where at least 10 infants were positive for a given crAss-like phage cluster. Presence of cluster Delta 07

(Bäckhed $p = 0.009$, Yassour $p = 1$, combined $p = 0.007$, Fisher's exact test) and *any* crAss-like phage cluster (Bäckhed $p = 0.004$, Yassour $p = 0.32$, combined $p = 0.001$, Fisher's exact test) was significantly associated with vaginal delivery (Supplementary Data 1). *P*-values are uncorrected for multiple hypothesis testing. No significant associations between crAss-like phage presence and breastfeeding status were found.

Although a crAss-like phage similar to cluster Beta 06 phages was recently cultured on a *Bacteroides intestinalis* host[20], the hosts of other crAss-like phages have yet to be identified. We searched for bacterial taxa that were differentially abundant between crAss-like phage positive and negative infants to make inferences about potential hosts. Samples from vaginally born infants at three or four months of age were included to allow comparisons across the two studies at a similar time point. Bacterial relative abundances were transformed to centered log-ratios, and differential abundance was calculated with the R package ALDEx2[33] (see Methods). *P*-values were calculated with the two-sided Wilcoxon rank-sum test and corrected for multiple hypothesis testing[34]. Due to limited sample numbers, we considered presence of any crAss-like phage as a group.

At the genus level, Collinsella was the most enriched taxon in infants positive for any crAss-like phage (corrected $p = 0.0167$, two-sided Wilcoxon test) (Supplementary Fig. 6, Supplementary Data 6). Several members of the Collinsella genus, including *Collinsella aerofaciens* (corrected $p = 0.0239$, two-sided Wilcoxon test) were also the most enriched species in these infants. Certain species of the genus Bacteroides were also significantly enriched to a lesser degree, such as *Bacteroides massiliensis* (corrected $p = 0.0499$, two-sided Wilcoxon test). Collinsella is a member of Actinobacteria, an entirely different phylum than the posited Bacteroidetes hosts[4]. Collinsella was previously identified as a signature of the developing anaerobic infant microbiome[15], but further work is necessary to determine if these species have a direct or indirect influence on the acquisition of crAss-like phages.

Next, we searched for putative mother-infant transmission of crAss-like phages. Depending on the cluster, 0–16 families (median 2.5, SD = 5.1) have the same crAss-like phage cluster detected in at least one mother and matched infant sample (Supplementary Data 5). We extended the metagenomic assembly analysis above, using a pangenome compiled from all crAss-like genomes in each cluster in place of the p-crAssphage reference. We find cases where mothers and infants share an assembled genome with >99.7% identity and at least 20 kb of aligned sequence in 7/10 clusters. As expected, candidate transmission events in cluster Alpha 01 matched p-crAssphage. Overall, we observe putative transmission in 21/42 (50%) cases where mother and infant have the same crAss-like phage cluster (Supplementary Data 5). In two families, we observed putative transmission of two separate crAss-like phage clusters. 50% is likely an underestimate of the true transmission rate, because many comparisons were limited by low sequencing depth and poorly assembled draft genomes. Interestingly, a subset of infants have a crAss-like phage but their mothers do not. This could be due to waxing/waning amounts of crAss-like phages in mothers, as has been described in adults[14]. If this is the case, the crAss-like phage may have been present in the mother at a level lower than our limit of detection and thus may have been transmitted to the baby. Alternatively, the baby may have acquired the crAss-like phage from an altogether different source, such as another housemate or environmental source.

**Similar p-crAssphage genomes found in FMT donors and recipients.** Another example of a perturbation where the gut microbiome, typically stable in adults, may acquire a large number of new microbes is when individuals experience infection with the gut pathogen *Clostridium difficile*, are treated with antibiotics, and subsequently receive fecal microbiota transplantation (FMT)[35,36]. Draper et al.[37] found that p-crAssphage relative abundance is decreased in individuals with recurrent *C. difficile* infection and that p-crAssphage could be transplanted from donor to recipient. However, strain-level transmission of p-crAssphage has not been explored in this patient population. We examined metagenomic sequencing data from Smillie et al.[35] and viral metagenomic sequencing data from Draper et al.[37] using the classification, assembly and comparison methods described above.

In the data from Smillie et al.[35], we detect p-crAssphage at ≥1x coverage in samples from two donors, MGH06D and MGH03D (Supplementary Data 1). 12 patients received stool preparations from either of those two donors. After FMT, 8/11 (73%) patients who received material from donor MGH03D were positive for p-crAssphage, while the individual who received material from donor MGH06D remained negative (Fig. 5a). Zero recipients who received FMT from a p-crAssphage-negative donor acquired p-crAssphage during the sampling period. We compared assembled p-crAssphage genomes from donors and recipients and found >99.8% identical sequences in samples from MGH03D and recipients of this donor's material, while samples from donor MGH06D had a distinct p-crAssphage sequence (96.7% identity to MGH03D) (Supplementary Fig. 7). Interestingly, individuals MGH11R and MGH12R experienced dynamic p-crAssphage presence, with the phage falling below and rising above the detection limit in subsequent samples. The assembled genomes remained highly similar in each case, suggesting a waxing/waning p-crAssphage population in individual MGH12R, who did not receive an additional FMT. In pre-FMT samples from recipients from this study, p-crAssphage was not detected, and only one sample was positive for a single crAss-like phage (Supplementary Data 1). This suggests the phages are substantially diminished in abundance when individuals are treated with drugs such as metronidazole, which has high activity against *Bacteroides species*.

Draper et al.[37] specifically sequenced the amplified viral content of the metagenome, so we adjusted the detection threshold to 10,000 reads (10x coverage) to reduce the number of false positives. P-crAssphage was detected in all 16 samples from donor D3 and 0/17 samples from donors D1 and D2. No pre-FMT samples from *C. difficile* colitis affected patients were positive for p-crAssphage or any crAss-like phage (Supplementary Data 1). In total, 7/7 patients who received FMT from donor D3 material became p-crAssphage positive; most remained positive for the 12 month duration of sampling (Fig. 5b). Assembled p-crAssphage genomes from donor D3 and the seven recipients had >99.5% nucleotide identity, suggesting colonization with the specific donor p-crAssphage strain (Supplementary Fig. 8). Patient P7 became p-crAssphage positive with a genome ~92% identical to the other donor and patients. P7 received material from donor D1, who was p-crAssphage negative, and therefore could have acquired the phage from a population below the detection limit in the donor or another source following re-establishment of host bacterial populations. Patient P13 had p-crAssphage present at ~40x in a single sample, but the assembled genome only had a total length of 12 kb and N50 of 2.8 kb. Samples with lower coverage assembled nearly complete p-crAssphage genomes with N50 > 60 kb. It is thus possible that P13's p-crAssphage detection is an artifact of PCR amplification that is often used in the sequencing of virus-enriched samples. These data show that p-crAssphage is frequently and efficiently transplanted via FMT and that p-crAssphage can stably engraft in FMT recipients for up to one year.

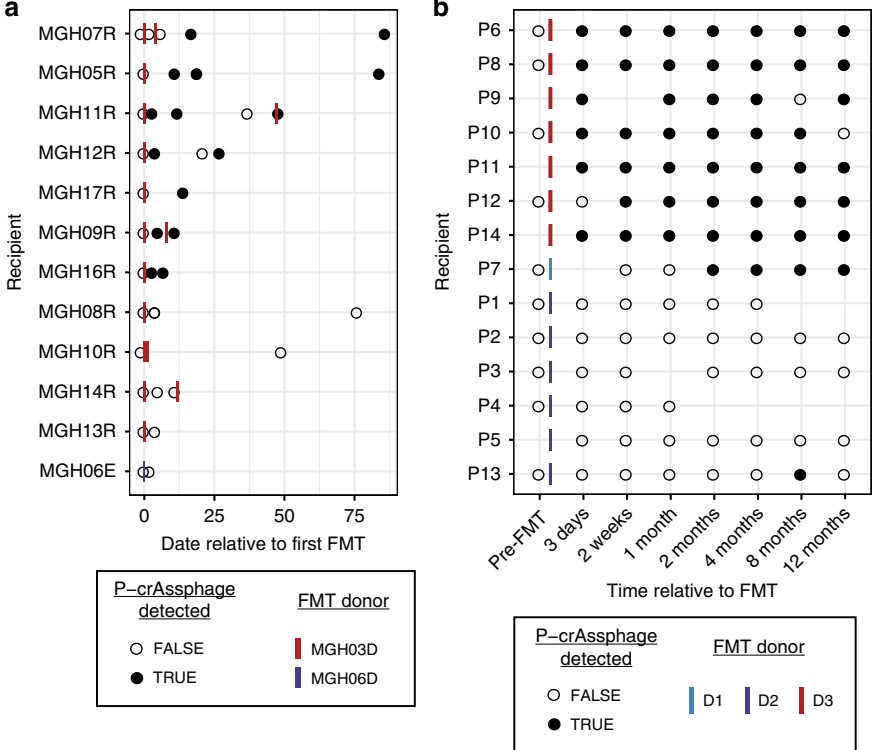

**Fig. 5 P-crAssphage status in patients receiving FMT over time. a** P-crAssphage detection at 1x coverage in samples from Smillie et al.[35] Both donors shown were p-crAssphage positive. Open circles represent a p-crAssphage negative samples and closed circles represent p-crAssphage positive samples. **b** P-crAssphage detection at 10x coverage in samples from Draper et al.[37] Donor D1 was p-crAssphage positive, while donors D2 and D3 were p-crAssphage negative.

## Discussion

The in silico discovery of p-crAssphage and recent publication of hundreds of crAss-like phage genomes has highlighted the diversity and global prevalence of these phages in human gut microbiomes[38]. CrAss-like phages have even been found in non-human primates[14], suggesting these phages have been evolving alongside humans for millions of years. However, it is currently unknown when and how an individual typically acquires crAss-like phages, as well as what level of strain diversity exists within the microbiome of an individual. The datasets examined here[15,16] contain mother-infant pairs sampled extensively during the first year of life and represent a unique opportunity to answer these questions.

We first characterized p-crAssphage and found no samples collected from infants within 24 h of birth met our 1x coverage threshold. P-crAssphage becomes increasingly prevalent as infants age, but does not reach the levels found in mothers by one year of life. The host(s) of p-crAssphage may not be present or have reached sufficient abundance in some infants by the end of sample collection. Infants acquire many of their gut bacteria through direct transmission from their mother, while gut viromes have been shown to remain unique between family members and twins[19]. In contrast to other members of the gut virome, we found nearly identical assembled p-crAssphage genomes in 6/10 cases where mothers and infants both harbor the phage, suggesting vertical transmission. However, we cannot rule out alternative possibilities, such as transmission from a different family member or from a common environmental source. We also observed cases where mothers and infants had unrelated p-crAssphage genomes and cases where infants had p-crAssphage but it was undetectable in the mother, which argue that the infants acquired p-crAssphage from an undetectable population in the mother or from another source.

It is currently unknown if individuals are typically colonized by a single or multiple p-crAssphage strains, or how similar or different these strains may be. We characterized the strain diversity of the p-crAssphage population in an individual by examining positions in the p-crAssphage genome where we detected multiple high-quality alleles. We found most mothers have a limited number of variable sites, with a median frequency of 0.04% across the 97-kb p-crAssphage genome, arguing that most mothers have a limited diversity of p-crAssphage strains. We did observe one mother with 100x more frequent variable sites, however. Infants generally have an order of magnitude fewer variable sites than mothers, suggesting a population that is further reduced in strain diversity, which may be the result of a bottleneck event upon acquisition or transmission. P-crAssphage is significantly less diverse than the second most abundant phage in these samples, Lactococcus phages 16802, where variable sites are detected with a median frequency of 5.9%. In cases where we observed putative mother-infant transmission, major alleles in the mother are primarily found in the infant, suggesting the mother's dominant strain is primarily responsible for colonizing the infant. The p-crAssphage population in infants develops additional variable sites over time, often at positions where only single alleles were detected in the mother. This could be due to the different bacterial hosts, nutritional sources and selective pressures in the infant microbiome, or simply due to random mutations.

In the most reductionist sense, two p-crAssphage genomes could differ at a single position and be considered different strains. However, we are most interested in strain variation that has functional consequences for the phage, its host or other members of the gut microbiome. Strain diversity in the p-crAssphage population of mothers is enriched for variants with predicted synonymous effects. However, we do observe enrichment for nonsynonymous (i.e. functional) variants in key genes,

including predicted tail fiber proteins. This suggests that there may be a benefit to maintaining nonsynonymous allelic diversity in these genes, such as the ability to infect a broader range of hosts. One isolated crAss-like phage[20] was noted to have a very specific host range, so variation in tail fiber genes may allow these phage to infect an increased range of bacteria. Laboratory experiments are necessary to further investigate this hypothesis, but could use exiting variation in the tail genes as a starting point to screen for expanded host range. P-crAssphage in infants has variable sites that are enriched for synonymous changes compared to mothers, but limited sample numbers made it difficult to determine enrichment for specific genes.

P-crAssphage is the first described member of a diverse group of crAss-like phages[5], with four "family" level and ten "genus" level classifications. Similar to p-crAssphage, we observe a trend of increasing prevalence with infant age for many clusters of crAss-like phages. Some clusters, such as Alpha 03, are prevalent in mothers but rarely or never observed in infants, suggesting the hosts of these phages have yet to reach sufficient abundance in the infant microbiome. We first observe a crAss-like phage at 1x coverage in samples collected three days after birth. In the case of family 633, the mother and three-day old infant have a Delta 07 phage with 99.3% alignment identity. Since we did not observe such early potential transmission events with p-crAssphage, this may represent the first detectable transmission of any crAss-like phage from mother to infant, and a lower limit for the time for a crAss-like phage to colonize the infant microbiome. Alternatively, crAss-like phages may not colonize the infant microbiome at such an early time, rather, they may be acquired through routes other than actual parturition. For example, the phages might be present in yet understudied niches, such as the mother's breast milk or the shared built environment of the baby and mother. Of note, *Bacteroides species*, which are posited to be the natural host of p-crAssphage and are known to be the host of a crAss-like phage[20] have previously been detected in breast milk[39]. Overall, we find nearly identical genomes in 50% of cases when we detect the same cluster crAss-like phage in both mother and infant, suggesting a transmission rate similar to p-crAssphage.

Regardless of the crAssphage status of the mother, we found a strong association of p-crAssphage and crAss-like phage presence with vaginal delivery, in contrast to what has been described previously[7]. One potential explanation is that vaginal birth is responsible for transmitting the phage from mother to infant. However, this is less likely in cases where infants harbor a phage undetected in the mother. Another possible explanation is that vaginal birth is responsible for seeding bacteria necessary for later colonization by crAss-like phages. Previous research found maternal seeding of bacteria from the class Bacteroidia was inhibited by C-section birth, supporting this hypothesis[24,40]. Future research with more balanced cohorts will likely clarify whether or not birth mode affects crAss-like phage acquisition and transmission. Unexpectedly, microbiomes of vaginally born infants positive for crAss-like phages were strongly enriched in *Collinsella species*. It is doubtful that this finding suggests new hosts for crAss-like phages, rather, Collinsella may be a hallmark for a developing and increasingly anaerobic infant microbiome that is capable of harboring these phages.

Finally, we observe that p-crAssphage is frequently transmitted via fecal microbiota transplantation (FMT) and can engraft stably in FMT recipients for up to one year. Engraftment of bacteria and phages has been well-studied in the case of FMT treatment for recurrent *Clostridium difficile* infection, and transplantation of p-crAssphage has been identified previously[35–37]. Our strain-level findings add new insight into the transmission of lytic bacteriophages. We assembled nearly identical genomes from both donors and recipients, highly suggestive of transmission of the specific p-crAssphage strain. Taken together, the results from both populations suggest that infants and patients receiving FMT have relatively unpopulated, naive microbiomes, providing an open niche for p-crAssphage to engraft into.

While this study suggests new principles about acquisition and transmission of crAss-like phages in the gut microbiome, it does have several limitations. First, we examined publicly available metagenomic data and were therefore limited to the available study cohort and sample size. In the mother-infant studies, stool samples were not collected from family members other than mothers, which could help determine other contributions to crAss-like phage acquisition in infants. Also, many infant birth samples were limited by low sequencing depth. As such, the estimates for acquisition and transmission presented here are likely underestimates. Sampling time points, processing techniques and study populations were different between the two studies, although both were conducted in Northern European individuals. Second, short-read sequencing data limited our ability to phase strain variants in the p-crAssphage genome. If the samples were resequenced with long-read sequencing approaches[41], we could obtain single reads spanning many variable sites. This would allow us to determine if the observed variants are the result of a smaller number of more divergent strain populations, or a high number of closely related strains. Finally, our group has become aware of false positive strain sharing results due to "barcode swapping" in dual-indexed Illumina sequencing libraries generated in our lab, which was first described in 2017[42]. As the indexing strategy was not reported for the public data we analyzed in this manuscript, we cannot be certain that the findings presented are not the result of this artifact. However, we believe our results, where only matched mother-infant pairs and matched FMT donor-recipient pairs share highly related crAss-like phage sequences, are unlikely to be explained by barcode swapping alone. The detrimental effect of barcode swapping also highlights the importance of reporting index sequences as a key part of making data publicly available.

Future work expanding on our findings should be directed towards answering several important questions. How stable are crAss-like phages transmitted from mother to infant over time? Are they lifelong inhabitants that, barring heavy antibiotic use, can be transmitted for generations? Are there exclusion principles that prevent the acquisition of a second, more divergent p-crAssphage strain? Additionally, our strain diversity analysis focused on p-crAssphage, but a wealth of diversity is also present in crAss-like phages. Better genome annotations and more concrete principles surrounding the identity and taxonomy of crAss-like phages will enable this research, and isolating, culturing and characterizing new crAss-like phages is a key next step. Finally, long-read metagenomic sequencing will enable better analysis of the strain populations among crAss-like phages in the mixed community of the microbiome. The ubiquity, distinct genome composition and ease of computational analysis with crAss-like phages may render them useful models for querying microbial transmission more broadly. Future work remains to determine precisely whether, and how, crAss-like phages influence the gut ecosystem and ultimately human health.

## Methods

**Sequence read preprocessing**. Raw sequencing reads from Bäckhed et al.[15] and Yassour et al.[16] were downloaded from SRA from each sample and preprocessed in a consistent way: TrimGalore version 0.5.0[43] was used to perform quality and adapter trimming with the flags "–clip_R1 15–clip_R2 15–length 60". SeqKit version 0.9.1[44] was used to remove duplicates with the command "seqkit rmdup–by–seq". Reads were mapped against the human genome using BWA version 0.7.17-r1188[45] and only unmapped reads were retained. Many infant samples at birth had low read counts after preprocessing, and samples with fewer

than 10,000 reads were removed from all subsequent analyses. This left 135 families with sufficient depth in at least one sample from mother and infant.

**Kraken2 classification.** For classification of p-crAssphage, we built a Kraken2[21] database containing all bacteria, viral and fungal genomes in NCBI GenBank assembled to complete genome, chromosome or scaffold quality as of February 2019. Human and mouse reference genomes were also included in the database. A Bracken[46] database was also built with a read length of 150 and k-mer length of 35. P-crAssphage is represented by a 97-kb genome (accession NC_024711.1). Multiple crAss-like phages are present in GenBank and would cause reads mapping to multiple genomes to be classified at the least common ancestor of "crAss-like viruses." To prevent this from happening, other crAss-like genomes were removed from the database.

For classification of crAss-like phages, we added to the viral database, replacing the original "crAss-like viruses" clade with genomes in the proposed subfamily and cluster hierarchy described in Guerin et al.[5] Kraken2 was used with default classification parameters on paired-end reads.

For testing associations between crAss-like phage presence and other bacterial taxa, reads were classified by using Kraken2 with default parameters on paired-end reads, and Bracken was used for abundance estimation with the parameters "-r 150 -l S -t 10".

**Assembling and comparing crAss-like phage genomes.** Preprocessed sequencing reads were assembled with SPAdes version 3.13.1[47] using the '–meta' flag. Contigs ≥500 bp were aligned with BWA against either the p-crAsspahge reference genome or composite genomes from all the crAss-like phages in a cluster from Guerin et al.[5] Resulting contigs were assessed for their N50 and total assembly length. Pairwise comparisons were conducted with nucmer version 4.0.0beta2[48] and the average identity and total length of 1–1 aligned segments was reported. The heatmap in Fig. 1 was clustered on the euclidean distance between samples with the ward.d2 clustering method and plotted with the *heatmap.2* function in the *gplots* package for R[49].

**SNPs and multiallelic sites.** SNPs were called with Snippy[50] with freebayes[51] as the variant caller using the p-crAssphage reference at sites covered ≥10x. Filtering, decomposition and normalization of variants was necessary to compare between samples and was conducted with vt version 0.5[52] and bcftools version 1.9[53]. The output of snippy, *snps.raw.vcf*, was used in this command: "vt decompose -s snps. raw.vcf | vt decompose_blocksub -a - | bcftools norm -f crassphage_reference.fasta -m -any | bcftools view–include 'QUAL ≥ 100 && FMT/DP ≥ 10 && (FMT/AO)/ (FMT/DP) ≥ 0'". We calculated the transition/transversion ratio of detected variants using *vcftools*[54] version 0.1.16. Considering all detected variants agnostic of samples, the transition/transversion ratio is 2.91 for all SNPs, 2.41 for fixed SNPs after variant decomposition and 2.42 for multiallelic SNPs at >0.1 AF after variant decomposition. Considering samples individually, the median Ts/Tv ratios are 3.40, 2.80 and 3.23 (SD = 2.3, 1.5, 3.4), respectively. The median sample numbers are higher because of samples with few detected transversions producing a comparatively high ratio.

To compare fixed SNPs between samples, we only consider sites covered ≥10x in both samples. The reported SNP % identity is 1 - (the number of fixed SNPs different between samples)/number of sites covered ≥10x in both samples).

Multiallelic sites were called as sites with two alleles and ≥5 reads supporting each allele. We report a normalized statistic ($F_{multi}$) to compare multiallelic sites across samples with highly variable coverage. At a given minor allele fraction(AF), $F_{multi}$ is the proportion of multiallelic sites with a minor AF > $x$ among those sites covered well enough to detect a minor AF of $x$. Effects of multiallelic variants were predicted with SnpEff version 4.3[27] using the p-crAssphage genome annotation available on GenBank and the flags "ann -noLog -noStats -no-downstream -no-upstream -no-utr -t". When mothers had multiple samples, we used the one with the highest p-crAssphage coverage for multiallelic site analysis.

**CrAss-like phage correlation with bacterial abundance.** We used the outputs of Bracken to test for differential abundances in taxa between groups. Matrices of reads classified to each taxon in each sample were filtered to keep only taxa with an abundance of at least 0.001 and nonzero values in at least 30% of samples. Zeros in the data were replaced with the Geometric Bayesian multiplicative method in the zCompositions version 1.3.2–1 package for R[55]. Differential abundance between groups was calculated with the ALDEx2 version 1.16.0 package for R[33].

**FMT data analysis.** Data from the FMT studies[35,37] were processed, assembled and compared in the same way as mother/infant data. Sample MGH06R[35] was excluded from the FMT cohort analysis as it could not be definitively determined whether sample designated as pre-FMT was actually collected prior to transplantation (personal communication with authors).

**Reporting summary.** Further information on research design is available in the Nature Research Reporting Summary linked to this article.

## Data availability

All data analyzed in this study are publicly available[15,16,35,37]. We have deposited the p-crAssphage and crAss-like phage assembled genomes as Supplementary Data 7 along with this manuscript.

## Code availability

Workflows for preprocessing, metagenomic assembly, assembly comparison and SNP calling in metagenomic data were used in this manuscript and can be found at: https://github.com/bhattlab/bhattlab_workflows.

A workflow for Kraken2 classification can be found at: https://github.com/bhattlab/kraken2_classification.

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

## Acknowledgements

We thank Benjamin Good for advice on statistical tests related to Fig. 3, and Dylan Maghini for helpful comments on the manuscript. This work was supported in part by the National Science Foundation Graduate Research Fellowship and the Stanford CEHG Pre-Doctoral Fellowship (FBT) as well as the National Cancer Institute NIH K08 CA184420, the Damon Runyon Clinical Investigator Award, and the Amy Strelzer Manasevit Award (ASB). This work was supported in part by the NIH grant P30 CA124435 which supports the following Stanford Cancer Institute Shared Resource: the Genetics Bioinformatics Service Center.

## Author contributions

B.A.S. and F.B.T. performed computational analysis. G.S. designed the multiallelic site analysis and edited the manuscript. B.A.S., F.B.T., and A.S.B. designed the study and wrote and edited the manuscript.

## Competing interests

The authors declare no competing interests.
