## [Peer Review File · Nature Communications]

Reviewers' comments:

Reviewer #1 (Remarks to the Author):

The manuscript "Transmission & Persistence of crAssphage, a Ubiquitous Human-Associated Bacteriophage" by Tamburini et al is a nice piece of work that utilizes both existing public datasets and a newly generated dataset to evaluate the host variability, transmission, and persistence of the crAssphage bacteriophage. The authors utilized shotgun metagenomics datasets and techniques, with a focus on SNV analyses and alignments, to answer their questions. The authors' observations provide additional insight into crAssphage, which is a ubiquitous phage of great interest and importance to the field. These insights are relevant because they setup future work, both from the authors and other scientists who read the paper, that will continue to elucidate the underlying biology of this system.

Overall I enjoyed reading this paper. The data was interesting and presented well, and the overall organization was easy to follow. I did note some areas of recommended clarification and/or revision for the authors.

1. The authors mention that the dataset they generated will be deposited into the NCBI SRA at the time of publication, which unfortunately means we reviewers are unable to review the quality of that data or deposition, including the completeness of the study metadata that is made publicly available. I recommend the authors at least make their deposition available to the reviewers, regardless of whether it is made public or not.

2. Starting at line 48 the authors describe an analysis in which they estimate the degree of crAssphage strain variation within human gut microbial communities using SNV quantification. The authors link the frequency of the multi allelic sites to the degree of strain variability, but I was left wondering how this compares to other viruses in those metagenomes? Are the crAssphage numbers uniquely low or is this also observed in other phages within those metagenomic datasets? This is a comparison that could be a control and help provide some context to the reader. It would be helpful if the authors included histograms, as presented in figure 1, for other viruses so that we can determine whether this is an observation uniquely associated with crAssphage within the datasets.

3. The authors discuss the implications of crAssphage detection in their sample sets, but it seems as though this could be impacted by sequencing depth of the samples. It was unclear whether the authors controlled for sequencing depth in their analyses by using a technique such as subsampling? How variable was the sequencing depth in the sample sets? The authors should clarify this point in the text and mention the implications of depth and correction in the findings.

4. Could the authors clarify the QC measures they took around their SNV calling process? Specifically I am thinking about the transition/transversion ratio of the SNVs that were called, as a way to ensure the results are not merely PCR artifacts? I know that it is difficult to compare it to existing values for QC since it is such a new and exploratory space, but it would be helpful to see how they compare across studies and to confirm that they are biologically relevant across the datasets. This would also be a nice reference for future studies that perform a similar analysis on this phage.

5. This is a minor point, but I recommend the authors go through and confirm consistent use of either past or present tense in the manuscript. For example around line 148, the tense was switching between past and present.

6. Another recommendation is that the authors include more text at the end to describe how the work will inform future studies. What do the authors plan to do with the information, and what do they hope others will take away from the work? What hypotheses were generated from the study

and how can they be followed up on? The authors do hint at this through the paper, but it could be nice to really lay it out for the readers at the end. This point is a suggestion, and the authors are free to take it or leave it.

Reviewer #2 (Remarks to the Author):

Tamburini and colleagues present information on the persistence and transmission of CrAssphage, a seemingly ubiquitous member of the human gut microbiota. They utilize public and study-generated whole community metagenomes to study the presence and absence of CrAssphage using read mapping. They show that CrAssphage can be vertically transmitted from mother to infant and can be acquired via faecal microbiota transplantation and in immunocompromised patients undergoing hematopoietic cell transplantation. Once acquired the authors report that CrAssphage persists stably within individuals over time (days to months).

Overall, the paper is well written and is focused on a topic which is of great interest to many working within the field of the human gut microbiome. The statistical analysis conducted is sound and well chosen. There are, however, a number of major drawbacks as it stands.

- There is a general lack of detail in the presentation of results, discussion and methods – timelines of sampling, more detailed description of the public datasets and the datasets generated within the study (general sequence stats etc), lack of systematic evaluation of the results - that make it hard to follow and assess the paper in a comprehensive manner, leaving a number of open questions.
- Is there anything interesting about the datasets that show the transmission pattern – clinically, geographically etc...
- I really miss the ecological context of the transmission. Which phage types are being transmitted alongside CrAssphage? Potential hosts?
- What is the relative abundance of the CrAssphage types within the donor-recipient or mother-infant pairs?
- Could you reconstruct the genomes of the CrAssphage that you detect in the datasets?
- Figure 2 – need to indicate that datasets are taken at different timepoints for each recipient of FMT.
- HCT/Figure 3 - longitudinal samples are mentioned – over 4 months - but I have no idea when samples were taken and how often - just the room occupancy.
- The evidence for an environmental source of CrAssphage in the context of the HCT patients sampled here is scant and warrants further more developed analysis.
- The Discussion of the results generated is very thin, with a lack of reference to ongoing efforts in this domain, e.g. Draper et al., 2018 Microbiome 2018 –maybe published after the authors submitted their paper(?), but also in more general terms.

General comments in response to Reviewers 1 & 2:

Thank you for your helpful guidance. In response to both the reviewers' comments, we carried out additional analyses, that led to several important changes in the manuscript. To that end, we would like to raise a few points to both reviewers with respect to the revised manuscript.

The manuscript has changed significantly since the first submission, which has addressed many of the reviewer comments and made some of them no longer relevant as the results they relate to are not included in this revised manuscript.

- 1) The first author has changed due to Dr. Fiona Tamburini graduating and Benjamin Siranosian taking over the project.
- 2) We have discovered issues in the metagenomic sequencing generated in our lab from HCT patients that invalidates the results as previously presented. These are not the result of mishandling; rather, they are the result of "barcode swapping", which was initially described in 2017 (<https://www.illumina.com/science/education/minimizing-index-hopping.html>) and which we did not account for in our initial analysis. This is detailed below:
 - Dual-indexed DNA sequencing libraries were prepared with the Illumina Nextera Flex or Nextera XT kits. Occasionally, a sample had a high abundance of crAssphage shared one barcode sequence with a crAssphage negative sample. In these cases, even barcode swapping at a rate of fractions of a percent resulted in thousands of crAssphage reads in the truly negative sample. We were then able to assemble identical crAssphage genomes from each sample, leading to a false positive finding. All crAssphage sharing events reported in the previous version could be explained by barcode swapping.
 - As such, we have removed the piece about HCT patient transmission of crAssphage. As the manuscript now analyzes entirely publicly available metagenomic sequencing data, no SRA submission will be made.
 - As the indexing strategy was not reported for the public data we analyzed in this manuscript, we cannot be certain that the findings presented are not the result of barcode swapping. However, we believe our results, where only matched mother-infant pairs and matched FMT donor-recipient pairs share highly related crAss-like phage sequences, are unlikely to be explained by barcode swapping alone.
- 3) Additional analyses have been added, and additional public datasets have been considered.

Reviewer #1 (Remarks to the Author):

The manuscript "Transmission & Persistence of crAssphage, a Ubiquitous Human-Associated Bacteriophage" by Tamburini et al is a nice piece of work that utilizes both existing public datasets and a newly generated dataset to evaluate the host variability, transmission, and

persistence of the crAssphage bacteriophage. The authors utilized shotgun metagenomics datasets and techniques, with a focus on SNV analyses and alignments, to answer their questions. The authors' observations provide additional insight into crAssphage, which is a ubiquitous phage of great interest and importance to the field. These insights are relevant because they setup future work, both from the authors and other scientists who read the paper, that will continue to elucidate the underlying biology of this system.

Overall I enjoyed reading this paper. The data was interesting and presented well, and the overall organization was easy to follow. I did note some areas of recommended clarification and/or revision for the authors.

We thank the reviewer for spending time to carefully review our paper and for their enthusiasm about the overall findings, general interest and organization of the manuscript. We thank the reviewer for the suggestions on how to improve/clarify our manuscript, and have incorporated all of the suggestions outlined below, as noted in the point-by-point response to the reviewer's comments. We are thankful for the feedback and believe this has substantially strengthened our manuscript.

1. The authors mention that the dataset they generated will be deposited into the NCBI SRA at the time of publication, which unfortunately means we reviewers are unable to review the quality of that data or deposition, including the completeness of the study metadata that is made publicly available. I recommend the authors at least make their deposition available to the reviewers, regardless of whether it is made public or not.

As described in the paragraph to the editor above, we discovered HCT patient sharing of crAssphage could be entirely explained by barcode swapping in our multiplexed illumina sequencing data. We are no longer including any new data generated from our lab, so no NCBI SRA deposition will be made.

2. Starting at line 48 the authors describe an analysis in which they estimate the degree of crAssphage strain variation within human gut microbial communities using SNV quantification. The authors link the frequency of the multi allelic sites to the degree of strain variability, but I was left wondering how this compares to other viruses in those metagenomes? Are the crAssphage numbers uniquely low or is this also observed in other phages within those metagenomic datasets? This is a comparison that could be a control and help provide some context to the reader. It would be helpful if the authors included histograms, as presented in figure 1, for other viruses so that we can determine whether this is an observation uniquely associated with crAssphage within the datasets.

We thank the reviewer for suggesting the inclusion of additional phages into our analysis; we have followed this suggestion with the following approach and findings.

We have classified sequencing reads from each mother-infant dataset with a Kraken2 database of all bacterial, fungal and viral genomes of scaffold, chromosome or complete genome quality

available in NCBI genbank as of February 2019. The crAss-like phage genomes produced by Guerin et al. (2018) were also included in this database. In total, the database contains 13,234 viruses at the species level.

CrAss-like phages are the most abundant group of phages. Lactococcus phages are the second most abundant group of phages, and the only other phages detected at $\geq 1x$ coverage in ≥ 10 samples from mothers and infants. The most abundant member of this group, *Lactococcus phage 16082*, is present in 34 samples at $\geq 1x$ coverage. We detect more multiallelic sites on average than p-crAssphage, with a median of 58.8 sites per kb at AF > 0.1 , corresponding to an increased level of strain diversity. Similar to p-crAssphage, this diversity could be the result of many closely related strains, or a smaller number of more divergent strains. The distribution of multiallelic site counts for this phage is reproduced below and in Figure S9, and discussed in the results section.

3. The authors discuss the implications of crAssphage detection in their sample sets, but it seems as though this could be impacted by sequencing depth of the samples. It was unclear

whether the authors controlled for sequencing depth in their analyses by using a technique such as subsampling? How variable was the sequencing depth in the sample sets? The authors should clarify this point in the text and mention the implications of depth and correction in the findings.

Thank you for pointing out this important consideration. We agree with the reviewer that depth of sequencing does impact the classification and detection of crAss-like phages. However, for the analyses presented in this paper, we believe subsampling the data while reasonable to consider, would further limit the sensitivity of and therefore be detrimental to downstream analyses.

- Many samples from infants collected within 24h of birth have low read counts. We have filtered out samples with fewer than 10,000 reads after preprocessing (as stated in the methods section).
- Subsampling (rarefaction) to the depth of the lowest sample would significantly reduce our ability to conduct analyses like genome assembly and SNP calling.
- We have selected the 1,000 reads (roughly 1x coverage) threshold as a value where metagenomic assembly and SNP calling starts to become viable. Certainly different thresholds would give different results, either by including lower coverage samples that we missed, or excluding samples on the low end of coverage.
- However, we are most interested that the phage genome is present and that we can assemble and call variants on it. Even in the analysis where we compare relative abundances of bacteria between infants, we believe rarefaction would limit sensitivity by discarding useful data, as has been suggested by Susan Holmes, an expert in the field of microbiome sequencing and rarefaction (McMurdie and Holmes, 2014).

To make the values more clear to the reader, we have included supplementary tables that show the number of sequencing reads in each sample, as well as the number of reads classified to each crAss-like phage cluster. We have also included summary information in the main text.

4. Could the authors clarify the QC measures they took around their SNV calling process? Specifically I am thinking about the transition/transversion ratio of the SNVs that were called, as a way to ensure the results are not merely PCR artifacts? I know that it is difficult to compare it to existing values for QC since it is such a new and exploratory space, but it would be helpful to see how they compare across studies and to confirm that they are biologically relevant across the datasets. This would also be a nice reference for future studies that perform a similar analysis on this phage.

We thank the reviewers for this suggestion and have clarified the methods section of our manuscript. Specifically, we are only working with high-quality SNPs for both the fixed-SNP transmission analysis and the multiallelic site analysis. The filtering commands are listed in the methods, and summarized here: SNPs were called with Snippy (Seeman 2015) with freebayes (Garrison and G. 2012) as the variant caller using the p-crAssphage reference at sites covered

≥10x. SNPs with QUAL scores <100 are filtered out (this corresponds to phred 100, ie 1e-10 probability that the call is wrong). Multiallelic sites required ≥50x coverage. These steps ensure the variant calls are correct given correct sequencing data, but do not account for potential PCR artifacts.

We have calculated the transition/transversion ratio (Ts/Tv) for the set of fixed and multiallelic SNPs for each sample in the mother-infant collection and reported the values in the methods section. Considering all detected variants agnostic of samples, the Ts/Tv ratio is 2.91 for all SNPs, 2.41 for fixed SNPs after variant decomposition and 2.42 for multiallelic SNPs at >0.1 AF after variant decomposition. Considering samples individually, the median Ts/Tv ratios are 3.40, 2.80 and 3.23, respectively. The median sample numbers are higher because of samples with few detected transversions producing a comparatively high ratio.

The reviewer is correct that this is a new and exploratory space and not many reference values exist in the literature. The one other manuscript with a comparable value is a preprint (Brown et al. 2019), where the authors assembled a novel crAssphage. They report a Ts/Tv value of 4.71 from 40 detected SNPs. Their estimate is higher than the majority of samples in our mother-infant collection, but also may be biased by the small number of detected mutations (40 vs >1200 on average in the mother-infant collection).

5. This is a minor point, but I recommend the authors go through and confirm consistent use of either past or present tense in the manuscript. For example around line 148, the tense was switching between past and present.

We thank the reviewer for noting this inconsistency. We have carefully reviewed the manuscript and now use a consistent tense.

6. Another recommendation is that the authors include more text at the end to describe how the work will inform future studies. What do the authors plan to do with the information, and what do they hope others will take away from the work? What hypotheses were generated from the study and how can they be followed up on? The authors do hint at this through the paper, but it could be nice to really lay it out for the readers at the end. This point is a suggestion, and the authors are free to take it or leave it.

We agree that the manuscript would be more helpful to readers if we explicitly, instead of implicitly, provide the take-away points and provide a sense of what hypotheses were generated from this paper and how they may be followed up upon. We have updated the last section of the manuscript to reflect this and have added a short paragraph that specifically touches upon hypotheses/future experiments and analysis.

Reviewer #2 (Remarks to the Author):

Tamburini and colleagues present information on the persistence and transmission of CrAssphage, a seemingly ubiquitous member of the human gut microbiota. They utilize public and study-generated whole community metagenomes to study the presence and absence of CrAssphage using read mapping. They show that CrAssphage can be vertically transmitted from mother to infant and can be acquired via faecal microbiota transplantation and in immunocompromised patients undergoing hematopoietic cell transplantation. Once acquired the authors report that CrAssphage persists stably within individuals over time (days to months).

Overall, the paper is well written and is focused on a topic which is of great interest to many working within the field of the human gut microbiome. The statistical analysis conducted is sound and well chosen. There are, however, a number of major drawbacks as it stands.

We thank the reviewer for their careful review of our work and appreciate their perspective that the paper is “well written” and is “focused on a topic which is of great interest to many” working within our field. We are also pleased and highly reassured that the reviewer agrees that our statistical analyses are sound and well chosen. We have addressed the areas for improvement in the manuscript and address them point-by-point below.

- There is a general lack of detail in the presentation of results, discussion and methods – timelines of sampling, more detailed description of the public datasets and the datasets generated within the study (general sequence stats etc), lack of systematic evaluation of the results - that make it hard to follow and assess the paper in a comprehensive manner, leaving a number of open questions.

We thank the reviewer for their feedback. We have added more detailed descriptions of these datasets in the main text to aid interpretation, and included a detailed table in supplement that contains all sample characteristics.

- Is there anything interesting about the datasets that show the transmission pattern – clinically, geographically etc...

We agree that this is a pertinent question and one of great interest. The two primary datasets we've analyzed (Yassour and Bäckhed) were both conducted on Northern European individuals with no additional geographic information presented. We did find significant associations of crAss-like phage presence with vaginal delivery, as has been noted in the text.

- I really miss the ecological context of the transmission. Which phage types are being transmitted alongside CrAssphage? Potential hosts?

We agree this is an interesting question, and searching for microbes transmitted along with crAss-like phages is an excellent way to search for potential hosts. This analysis is included in the main text, Figure S6 and Table S6.

- What is the relative abundance of the CrAssphage types within the donor-recipient or mother-infant pairs?

This information has been included with the characteristics of all samples analyzed in Table S1.

- Could you reconstruct the genomes of the CrAssphage that you detect in the datasets?

Yes, in cases where there is enough coverage to generate a contiguous assembly. We have included the assembly statistics for p-crAssphage and crAss-like phages in Table S2. We are also including all assembled genome sequences from the mother-infant datasets as a supplementary file.

- Figure 2 – need to indicate that datasets are taken at different timepoints for each recipient of FMT.

The FMT information is now presented in Figure 5. The x-axis has been labeled to reflect that time is relative to the first FMT for each patient.

- HCT/Figure 3 - longitudinal samples are mentioned – over 4 months - but I have no idea when samples were taken and how often - just the room occupancy.

As described in the comment to the editor above, we discovered HCT patient sharing of crAssphage could be entirely explained by barcode swapping in the multiplexed illumina sequencing. We are no longer including any HCT patient data in the manuscript.

- The evidence for an environmental source of CrAssphage in the context of the HCT patients sampled here is scant and warrants further more developed analysis.

As described comment to the editor above, we discovered HCT patient sharing of crAssphage could be entirely explained by barcode swapping in the multiplexed illumina sequencing. We are no longer including any HCT patient data in the manuscript.

- The Discussion of the results generated is very thin, with a lack of reference to ongoing efforts in this domain, e.g. Draper et al., 2018 Microbiome 2018 –maybe published after the authors submitted their paper(?), but also in more general terms.

We thank the reviewer for encouraging a more developed discussion, which was also suggested by Reviewer 1. Given the limitations of the letter format, our previous version was more heavily skewed to the results section. We have now addressed this in several ways. First, we adapted our manuscript to an article format, which allows for a much more developed discussion.

As in our response to reviewer 1's comments, we have now expounded upon our interpretation of results and provided a clear set of implications of these results, hypotheses that are

generated, and future potential experiments/analyses that should be carried out. We have also cited the manuscripts noted by the reviewer, as well as additional manuscripts that have been published on the crAssphage topic since the time of the original submission.

REVIEWERS' COMMENTS:

Reviewer #1 (Remarks to the Author):

It is great to see the revised manuscript. It demonstrates a major improvement from the previous version.

I commend the authors for addressing the issue around barcode swapping in their dataset after the initial submission. That is never an easy issue to deal with, but it looks like they addressed it well.

Overall they addressed the reviewer comments.

Reviewer #2 (Remarks to the Author):

The authors of the revised manuscript 'Acquisition, transmission and strain diversity of human gut-colonizing crAss-like phages' have delivered a really nice paper that will be of great interest to the field.

All my original comments and queries have been addressed thoroughly.

I really enjoyed reading this revised version and appreciate the attention to detail as well as the transparent nature of the development of this piece of work. Raising the issue (again) of barcode swapping of multiplexed Illumina data within this context further highlights what could be an important consideration for further study of crAss phages and the dynamics of the human gut virome in general.

One minor comment is that a discussion of the transmission of Lactococcus phages is largely missing. Inclusion of a few lines regarding this observation may be useful for the reader, but not a necessity.

Response to reviewers comments

Reviewer #1 (Remarks to the Author):

It is great to see the revised manuscript. It demonstrates a major improvement from the previous version.

I commend the authors for addressing the issue around barcode swapping in their dataset after the initial submission. That is never an easy issue to deal with, but it looks like they addressed it well.

Overall they addressed the reviewer comments.

We thank reviewer 1 for their helpful comments, which have enabled us to improve our manuscript.

Reviewer #2 (Remarks to the Author):

The authors of the revised manuscript 'Acquisition, transmission and strain diversity of human gut-colonizing crAss-like phages' have delivered a really nice paper that will be of great interest to the field.

All my original comments and queries have been addressed thoroughly.

I really enjoyed reading this revised version and appreciate the attention to detail as well as the transparent nature of the development of this piece of work. Raising the issue (again) of barcode swapping of multiplexed Illumina data within this context further highlights what could be an important consideration for further study of crAss phages and the dynamics of the human gut virome in general.

We thank reviewer 2 for their comments, and appreciate that their previous specific guidance has helped us improve our manuscript. We agree that barcode swapping is a major issue that is likely to impact future studies of the dynamics and transmission of phages, and we hope our transparent experience with this issue will serve as a useful example to other researchers.

One minor comment is that a discussion of the transmission of Lactococcus phages is largely missing. Inclusion of a few lines regarding this observation may be useful for the reader, but not a necessity.

We thank reviewer 1 for this suggestion; we have now included a brief section on transmission of Lactococcus phages (we do not detect any convincing evidence of transmission in these datasets).